# Examining the Effects of Anabolic–Androgenic Steroids on Repetitive Mild Traumatic Brain Injury (RmTBI) Outcomes in Adolescent Rats

**DOI:** 10.3390/brainsci10050258

**Published:** 2020-04-28

**Authors:** Jason Tabor, David. K. Wright, Jennaya Christensen, Akram Zamani, Reid Collins, Sandy R. Shultz, Richelle Mychasiuk

**Affiliations:** 1Department of Psychology, Alberta Children’s Hospital Research Institute, Hotchkiss Brain Institute, University of Calgary, Calgary, AB T2N 1N4, Canada; jbtabor@ucalgary.ca (J.T.); jennaya.christensen@monash.edu (J.C.); reid.collins@ucalgary.ca (R.C.); 2Department of Neuroscience, Central Clinical School, Monash University, Melbourne 3004, Australia; david.wright@monash.edu (D.K.W.); akram.zamani@monash.edu (A.Z.); sandy.shultz@monash.edu (S.R.S.)

**Keywords:** concussion, advanced MRI, diffusion, mRNA, development

## Abstract

Background: Repetitive mild traumatic brain injury (RmTBI) is increasingly common in adolescents. Anabolic–androgenic steroid (AAS) consumption among younger professional athletes is a significant risk factor for impaired neurodevelopment. Given the increased rates and overlapping symptomology of RmTBI and AAS use, we sought to investigate the behavioural and neuropathological outcomes associated with the AAS Metandienone (Met) and RmTBI on rats. Methods: Rats received either Met or placebo and were then administered RmTBIs or sham injuries, followed by a behavioural test battery. Post-mortem MRI was conducted to examine markers of brain integrity and qRT-PCR assessed mRNA expression of markers for neurodevelopment, neuroinflammation, stress responses, and repair processes. Results: Although AAS and RmTBI did not produce cumulative deficits, AAS use was associated with detrimental outcomes including changes to depression, aggression, and memory; prefrontal cortex (PFC) atrophy and amygdala (AMYG) enlargement; damaged white matter integrity in the corpus callosum; and altered mRNA expression in the PFC and AMYG. RmTBI affected general activity and contributed to PFC atrophy. Conclusions: Findings corroborate previous results indicating that RmTBI negatively impacts neurodevelopment but also demonstrates that AAS results in significant neuropathological insult to the developing brain.

## 1. Introduction

Mild traumatic brain injury (mTBI), such as concussion, is the result of head trauma due to impact or accelerative forces, producing shearing damage to axons and blood vessels within the brain [1]. In the U.S. alone, there are more than 1.1 million concussion-related visits to the emergency room each year, with adolescents being a highly vulnerable demographic [2]. Of these youth, many are high-performing athletes engaged in collision sports and are therefore at increased risk for sustaining repeated concussions [3]. While many athletes recover from concussion and return to play within 7–10 days, some suffer from persistent symptoms including cognitive deficits in memory, attention, and information processing, collectively known as persistent post-concussion symptoms (PPCS) [4]. Diffuse axonal injury from shearing forces may be responsible for many of the lingering symptoms identified after concussion [5]. Youth are hypothesized to be more susceptible to concussion and PPCS due to underlying factors such as continuing neuronal myelination [6], and an increased sensitivity to excitotoxic injuries [7].

In addition to their increased vulnerability to concussion, many adolescent athletes self-administer anabolic–androgenic steroids (AAS) in an effort to gain a competitive edge [8]. AAS are exogenous hormones including the male hormone testosterone or lab-made derivatives that bind to the androgen receptor [9]. Clinically, these drugs are used in hormone replacement therapy for hypogonadism [10], or chronic diseases such as HIV/AIDS [11]. There is significant AAS abuse in the U.S. adolescent population, with 3–4 million Americans having used AAS at least once in their lifetime [12]. Adolescents who consume AAS during puberty subject their hormonally sensitive brains to risks, as the addition of exogenous hormones at supraphysiological levels induces behavioural detriments and changes in neural connectivity [13]. Behavioural symptoms experienced by chronic AAS abusers include major mood syndromes such as mania and major depression [14], increases in hostility and aggression, and changes in anxiety [15].

As adolescents experience a rise in both repetitive mTBI (RmTBI) [16,17] and AAS use, there is a need to evaluate the cumulative effects these factors may have on neurodevelopment and post-concussive symptomology. Despite an abundance of literature regarding the neurological consequences of either RmTBI or AAS alone, there is a significant void with respect to how these factors interact in adolescent athletes. Therefore, this study aimed to investigate the potentially cumulative morphological, behavioural, and neuropathological outcomes of AAS and RmTBI by using the common AAS, Metandienone (Met) [18]. Male rats were administered Met or placebo, then randomly allocated to RmTBI or sham TBI groups. Following the injuries, a behavioural test battery was performed to assess for cumulative effects of Met and RmTBI on post-concussive symptomology. Brain morphology was examined with magnetic resonance imaging (MRI) using volumetric analysis of the prefrontal cortex (PFC), amygdala (AMYG), and corpus callosum (CC). Additionally, white matter integrity of the CC was examined through diffusion tensor imaging (DTI). These brain regions were selected as both AAS and RmTBI can negatively affect interconnected brain circuitry involved in cognitive performance, short-term working memory, stress responses, and fluctuations in socio-emotional dysregulation [19,20,21,22]. Changes to markers of neurodevelopment, neuroinflammation, stress responses, and repair processes were examined via changes in mRNA expression in the PFC and AMYG.

## 2. Materials and Methods

### 2.1. Subjects

All experiments were conducted in accordance with the Canadian Council of Animal Care and received approval from the University of Calgary Conjoint Faculties Research Ethics Approval Board (Ethics Approval Number: AC18-0069). Eighty-two male Sprague Dawley rats (Charles Rivers Laboratories) were housed in groups of 3 or 4 and maintained in an animal husbandry room at 21 °C with ad libitum access to food and water on a 12:12 hour light:dark cycle.

### 2.2. AAS Administration Protocol

Met (TripleBond, Guelph, ON, Canada) was administered continuously, beginning on P21 by dissolving the drug at a concentration of 1.5 mg/kg body weight/day in the animals’ drinking water. This dosage and route of administration were selected as clinical populations consume Met orally and at this dosage [18]. For the duration of the study (beginning at P21 and maintained until euthanasia), rats in the *Steroid* groups consumed Met while those in the *Placebo* groups received standard drinking water. Met administration began immediately after weaning, at P21, because we aimed to have the Met accumulate in the rats’ systems prior to the first TBI, which was inflicted at P41 (equivalent to adolescence in humans) [23].

### 2.3. RmTBI Procedure

Rats in each group were randomly assigned to receive 3 mTBIs or sham injuries with the lateral impact (LI) device as previously described [24]. Rats were lightly anesthetized with isoflurane and placed prone on a low-friction Teflon board. A 50 g weight was pneumatically propelled towards the animal’s head; average speed of 8.93 ± 0.17 m/s, inducing TBIs at ~81.5 G. The weight struck a protective “helmet”, which prevented structural damage to the skull, while still propelling the rat into a 180° rotation. Through the LI technique, the brain experiences acceleration/deceleration and rotational forces akin to typical sports-related concussion [24]. mTBIs and sham injuries were executed on P41, P44, and P47. This randomization generated 4 distinct groups; *Placebo* + *Sham*, *n =* 21; *Placebo* + *RmTBI*, *n* = 21; *Steroid* + *Sham*, *n* = 20; and *Steroid* + *RmTBI*, *n =* 20. Time-to-right (a measure for loss of consciousness) quantified the time each animal required to wake and flip from a supine to prone/standing position after injury.

### 2.4. Behavioural Testing

A behavioural test battery designed to measure post-concussive symptomology was performed for all 82 rats [24,25]. An *Open Field* paradigm [26] was employed to measure general locomotor activity on P49 or post-injury day 2 (PID2). On P50 (PID3), rats were tested on the *Elevated Plus Maze* (EPM), a behavioural test for general anxiety [26]. The *Dominance Tube* test was performed on PID4 to measure aggression levels [27]. Match pairings consisted of *Steroid* vs. *Placebo* rats, with *Sham* vs. *RmTBI*. Three trials per pairing were performed with time in the tube, trial wins, and win percentage recorded for each animal. A *Novel Context Mismatch* (NCM) test was employed to measure short-term working memory ability [28]. NCM training took place from P54-P56, with a probe trial of the test occurring on P57 (PID10). Lastly, depressive-like behaviours were tested in the *Forced Swim* paradigm [29] on PID14. 

### 2.5. mRNA Analysis

Animals were euthanized at PID15. Fifty rats were anesthetized via isoflurane inhalation, weighed, and decapitated. Using the Zilles atlas [30], brain tissue from both hemispheres of the PFC and AMYG was removed, flash-frozen with dry ice, then stored at −80 °C. Extraction of total RNA from each brain region was performed with the Allprep RNA/DNA Mini Kit using manufacturer protocols (Qiagen, Hilden, Germany), for molecular analysis. Sample purity and concentration were tested with a NanoDrop 2000 (Thermo Fisher Scientific, USA). Two micrograms of purified RNA was reverse transcribed into cDNA by employing the oligo(dT)20Superscript III First-Strand Synthesis Supermix Kit (Invitrogen, Carlsbad, CA, USA). 

Four genes were chosen, providing key information on the effects of RmTBI and AAS on neurodevelopment, neuroinflammation, stress responses, and repair processes. The genes selected were as follows: ionized calcium-binding adaptor molecule 1, (*Iba1*), brain-derived neurotrophic factor, (*Bdnf*), cAMP response-element binding protein (*Creb*), and glial fibrillary acidic protein (*Gfap*). *Iba1* is involved in microglial activation during the neuroinflammatory response, playing roles in the proliferation and migration of microglia to sites of neural damage to enact the appropriate immune response [31]. *Bdnf* is involved in proper neurodevelopment through regulation of neurogenesis and neural plasticity, both a part of proper learning and memory function [32]. Altered *Bdnf* levels have also been associated with depression [33]. *Creb* is a transcription factor susceptible to disruption through brain injury, resulting in potential changes to complex learning and memory mechanisms [34]. *Gfap* is found in glial cells throughout the brain and is a specific marker of brain injury [35]. 

qRT-PCR primers were designed in-house using Primer3 (http://bioinfo.ut.ee/primer3) and purchased through Integrated DNA Technologies (Coralville, IA, USA). Samples were run in duplicate on a 96-well plate, with each target gene processed. qRT-PCR was performed and analyzed with the Applied Biosystems™ StepOnePlus™ Real-Time PCR System (Thermo Fisher Scientific, USA). Two housekeeping genes, *Ywhaz* and *CycA* were used in the 2^−ΔΔCt^ technique to determine relative target gene expression [36].

### 2.6. MRI Analysis 

On P62, thirty-two rats were transcardially perfused with phosphate-buffered saline (PBS) followed by 4% paraformaldehyde (PFA) in PBS. Rats were decapitated, and brains were removed then stored in 4% PFA at 4 °C. Brains were washed in PBS and embedded in Agar for scanning as described previously [21]. Images were acquired using a 9.4T MRI (Bruker, Billerica, MA, USA) with actively decoupled volume transmit and 4-channel surface receive coils. A 3D multi-gradient echo image was acquired with the following imaging parameters: repetition time (TR) = 66 ms; 14 echoes with minimum echo time (TE) = 2.7 ms and echo spacing = 3.4 ms; field of view (FOV) = 25.6 × 19.52 × 12.8 mm^3^; matrix size = 160 × 122 × 80; and resolution = 0.16 × 0.16 × 0.16 mm^3^. A diffusion-weighted image was acquired with a 2D echo planar-based sequence and the following imaging parameters: TR = 5 s; TE = 45 ms; FOV = 24 × 24 mm^2^; matrix size = 96 × 96; resolution = 0.25 × 0.25 mm^2^; 48 slices with thickness = 0.25 mm; diffusion duration = 6 ms; diffusion separation = 18 ms; *b*-value = 4000 s/mm^2^; 81 directions; and 3 non-diffusion (b_0_) images.

Individual gradient echo images were averaged to improve signal to noise ratios, and 5 regions of interest (ROIs) were traced on each brain. ROIs included the CC and ipsilateral and contralateral PFC and AMYG. DTI metrics were calculated using MRtrix3 (www.mrtrix3.org). Advanced Normalization Tools (ANTs, http://stnava.github.io/ANTs/) registered the mean b0 image to the mean T2*-weighted image with standard symmetric normalization (SyN) [37]. The ROIs were transformed from the T2* image space to DTI space using the inverse transforms and the mean value for each calculated using MATLAB. Fractional anisotropy (FA), radial diffusivity (RD), axial diffusivity (AD), and the apparent diffusion coefficient (ADC) were obtained. 

### 2.7. Telomere Length Analysis

A sample of ear notch tissue was taken at time of death. Genomic DNA was extracted using the Sigma REDExtract N-AMP Tissue PCR kit. Quality and concentration were measured using the NanoDrop 2000 (Thermo Fisher Scientific, Waltham, MA, USA). Telomere length calculations used the telomere repeat number to single copy ratio (T/S) to investigate telomere length, with the single copy corresponding to the 36B4 gene as previously described by our laboratory [25].

### 2.8. Serum Hormone Analysis

On PID15, trunk blood was collected at euthanasia to measure testosterone. Samples were permitted to clot for 30 min at room temperature and were then centrifuged for 15 min at 1000 g at 4 °C. Serum was collected and stored at −80 °C. ELISAs for Testosterone (Abcam Inc, Canada) and Growth Hormone (Invitrogen, Carlsbad, CA, USA) were performed according to manufacturer protocols. Standards, controls, and samples were run in duplicate and measured using the BioTek Synergy H.T. plate reader along with Gen5 2.00.18 software. 

### 2.9. Statistical Analysis

Statistical analyses were performed with SPSS 23.0, and *p* ≤ 0.05 was considered statistically significant. Two-way ANOVAs with Treatment (Steroid vs. Placebo) and Injury (RmTBI vs. Sham) as factors were performed for each of the imaging, behavioural, and molecular outcomes. Post hoc (LSD) pairwise comparisons were carried out where appropriate. Error bars on graphs represent ± SEM. All data are available upon request from the corresponding author.

## 3. Results

### 3.1. Animal Characteristics

Body weight at the initial mTBI, along with brain weight and body mass index (BMI), was recorded at the time of euthanasia. The two-way ANOVA for BMI showed a main effect of Met treatment (F_(1,61)_ = 7.92, *p* < 0.01), where rats receiving Met treatment had higher BMIs compared to placebo-treated rats, which verifies that the Met treatment had an anabolic effect on the body. The two-way ANOVA of brain weight showed a main effect of Met (F_(1,81)_ = 4.92, *p* = 0.03), whereby Met-treated rats had increased brain weight at time of euthanasia.

### 3.2. MRI Volumetric Analysis

Figure 1 displays the results of the MRI volumetric analysis, and Table 1 shows all statistical results from the two-way ANOVAs for all MRI-based assessments. With respect to brain region volumes, RmTBI only decreased volume in the contralateral (CL) PFC, and steroids decreased brain volume in the ipsilateral (IL) and CL PFC and increased volume in the IL and CL AMYG. CC volume was not affected by RmTBI or steroids, and there were no significant RmTBIs by steroid interactions in any brain region.

### 3.3. DTI Analysis

Figure 2 displays the results of the DTI analysis. There were main effects of steroid treatment increasing AD, ADC, and FA values, but no significant effects on RD. Additionally, there were no main effects of injury on DTI metrics, nor any significant interactions between steroid treatment and RmTBI. 

### 3.4. Behavioural Testing

Figure 3 displays the results for the behavioural test battery performed on all animals. Table 2 contains statistical results from the two-way ANOVAs for the behavioural tests. Steroids affected 2/6 behavioural tests including the forced swim task (increasing time spent immobile, indicating increased depressive-like behaviour), and the NCM task (less time with the novel object, indicative of poorer short-term memory) in both RmTBI and sham animals. RmTBI impaired performance on 2/6 tasks (increased time-to-right and decreased activity in the open field). Additionally, there was a steroid by RmTBI interaction on the NCM task, with post-hoc analyses demonstrating that this was driven by sham animals, whereby steroid exposure significantly decreased performance on the NCM task (*p* < 0.01). This interaction in the NCM task was also driven by animals in the placebo group, whereby those who received RmTBI performed significantly worse than those in the sham group (*p* < 0.01). There was also a significant RmTBI by steroid treatment interaction in the dominance tube test, with post-hoc analyses showing that this was driven by RmTBI animals where the steroid group had significantly lower scores than the placebo group (*p* = 0.04)

### 3.5. mRNA Expression

mRNA expression was analyzed for four different genes (*Bdnf*, *Iba1*, *Creb,* and *Gfap*) in the PFC and AMYG, which is shown in Figure 4. See Table 3 for a detailed summary of statistical findings. The two-way ANOVA analysis of gene expression in the PFC showed 2/4 genes affected by steroids (decreases in *Creb* and increases in *Gfap* expression), while no genes in this brain area were influenced by RmTBI. There were no significant interactions in any genes analyzed in the PFC.

Steroids only affected one of four genes in the AMYG (decreases in *Bdnf* expression), and RmTBI also only affected one gene (decreases in *Gfap* expression). There were also significant interactions affecting gene expression in the AMYG. The two-way ANOVAs demonstrated a significant RmTBI by steroid treatment interaction influencing the expression of *Bdnf*, *Creb*, and *Gfap*. Post hoc analyses for *Bdnf* demonstrated that the interaction was driven by the sham animals, whereby steroid exposure decreased *Bdnf* expression when compared to animals receiving the placebo treatment (*p* < 0.01). This interaction in *Bdnf* was also driven by the placebo animals, whereby RmTBI decreased *Bdnf* expression when compared to shams (*p* < 0.01). For *Creb*, post hoc analysis demonstrated that the interaction was driven by sham animals, whereby steroid treatment decreased *Creb* expression when compared to placebo animals (*p* = 0.01), and by steroid animals, whereby RmTBI exposure increased *Creb* expression when compared to sham injuries (*p* = 0.01). For *Gfap*, post hoc analyses demonstrated that the interaction was driven by the following: (a) the steroid-treated animals, whereby RmTBI decreased *Gfap* expression compared to sham injuries (*p* < 0.01); (b) placebo animals, whereby RmTBI decreased *Gfap* expression compared to sham injuries (*p* < 0.01); and (c) by the sham animals, whereby steroid exposure decreased *Gfap* expression compared to placebo-treated animals (*p* = 0.01); see Figure 4.

### 3.6. Telomere Length

At PID15, skin samples from ear notches were collected and examined for changes in telomere length. The two-way ANOVA showed a main effect of RmTBI (F_(1,50)_ = 7.38, *p* < 0.01), whereby rats who had RmTBIs had shorter telomeres compared to rats with sham injuries. There were no significant effects of Met treatment nor any significant interactions (Figure 4).

### 3.7. Serum Testosterone Levels

Serum levels of testosterone were examined at euthanasia (PID15) to verify that the rats consumed adequate Met treatment, as AAS use results in a suppression of the hypothalamic–pituitary–gonadal axis, thereby reducing endogenous testosterone production [38]. The two-way ANOVA showed a main effect of Met treatment (F_(1,67)_ = 39.71, *p* <.01), whereby animals who were given Met had significantly lower serum testosterone levels. There were no significant effects of RmTBI, or any significant interactions (Figure 4).

### 3.8. Serum Growth Hormone Levels

Serum levels of growth hormone were examined at euthanasia (PID15) to investigate the potential for RmTBI to damage pituitary function [39] as well as the potential for Met treatment to modulate GH secretion [40]. The two-way ANOVA showed a main effect of RmTBI (F_(1,65)_ = 4.92, *p* = 0.04), whereby animals who received RmTBI had significantly elevated levels of GH compared to sham injury animals (Figure 4).

## 4. Discussion

AAS use and RmTBI during adolescence are associated with changes in depression, anxiety, social behaviour, and cognitive function [15,41]. As many symptoms of AAS abuse overlap with those of PPCS, we hypothesized that there would be cumulative impairments in brain morphology and behaviour after RmTBI and AAS treatment in rats. Although we did not see many cumulative effects, we did identify negative effects of AAS treatment on the adolescent brain. RmTBI alone affected behavioural measures of general activity, and loss of consciousness, while Met treatment affected measures of short-term working memory and depressive-like behaviour. Interestingly, there were also interactions between RmTBI and Met treatment on measures of short-term working memory and aggression.

Our MRI analysis (see Figure 1 and Table 1) found that the ipsilateral PFC had decreased volume due to Met treatment (both the *Steroid*+*RmTBI* and the *Steroid+Sham* exhibited reductions in volume when compared to the *Placebo+Sham* and *Placebo+RmTBI*). Conversely, when examining the contralateral PFC volume, we found reductions in response to both Met treatment, (both *Steroid+Sham* and *Steroid+RmTBI* had reduced volumes when compared to *Placebo+Sham* and *Placebo+RmTBI*, respectively) and RmTBI (the *Placebo+RmTBI* group had reduced volumes when compared to the *Placebo+Sham* groups, while the *Steroid+RmTBI* group also have reduced volumes when compared to the *Steroid+Sham* group). This supports clinical MRI literature showing that long term-AAS users have thinning of the frontal regions [22]. It is worth noting that in the contralateral PFC (Figure 1B), placebo animals that received RmTBI had similar volume decreases to those of steroid animals who received sham injuries, highlighting that chronic Met exposure alone was just as detrimental as multiple concussive injuries; interestingly, when combined, the Met treatment and RmTBI produced the lowest volumes in the contralateral PFC. Given that the PFC is involved in impulse control, social behaviour, personality, and cognitive function [42], decreased PFC volume provides evidence for the underlying neurobiology of violent behaviours and cognitive deficits displayed by long-term AAS users [15]. Our RmTBI-induced volumetric reduction is consistent with previous studies showing cortical thinning after mTBI [43]. It was unexpected that a lateral mTBI produced reductions in the contralateral PFC and not the ipsilateral PFC, contrary to a previous study by our laboratory using the LI technique, which found decreases in ipsilateral PFC after lateral RmTBIs (however, these results were found in females, while males did not show any reductions) [21]. This heterogenous volume reduction by RmTBI between the ipsilateral and contralateral PFC could be explained by a potential countercoup effect, as previously shown in other rodent mTBI models [44]. Although the external impact occurred on the left (ipsilateral) side of the head, the PFC may have been damaged from a sudden deceleration and impact to the right (contralateral) side of the skull.

Similarly, our increases in both the ipsilateral and contralateral AMYG volume in response to Met treatment are consistent with clinical studies of long-term AAS users [13]. Increased androgen exposure in the AMYG has been associated with threat reactivity and aggressive behaviour [45,46], which may have been reflected in our dominance tube findings. Figure 1D shows a similar pattern to Figure 1B, where the contralateral AMYG may have shown decreased volume by RmTBI due to a countercoup effect; however, this difference was not statistically significant.

DTI was performed to examine white matter pathologies in the CC (see Figure 2 and Table 1). Similar to previous studies, chronic AAS administration was found to increase AD and ADC, both of which are indicative of damaged white matter [47]. Curiously, we also found increased FA in the CC. Only one prior clinical study has found increased FA due to chronic AAS use [48]; however, the region examined was the inferior-fronto-occipital fasciculus. Despite these changes to white matter integrity, we did not see any volumetric changes in this brain region either. To our knowledge, our study is the first to employ DTI to investigate the effects of AAS in the CC. In summary, chronic AAS exposure was detrimental to white matter integrity in the CC, and this may predispose individuals to various neurodegenerative diseases such as Alzheimer’s disease or multiple sclerosis [49,50].

The behavioural test battery (see Figure 3 and Table 2) found that RmTBI alone increased loss of consciousness, which is in line with previous findings [19], and demonstrated that the animals did in fact experience mTBIs. While there were no cumulative effects of RmTBI and AAS on behaviour, we did demonstrate that chronic Met treatment severely impaired short-term working memory in sham rats in a similar manner to the placebo group who received RmTBIs. This is consistent with previous studies that have shown chronic AAS treatment can negatively affect memory in rodents [51]. Although we demonstrated significant impairments due to Met treatment and RmTBI, the NCM task may not have been sensitive enough to detect cumulative effects of Met treatment and RmTBI. 

Consistent with prior studies, Met treatment increased depressive-like behaviour. This may be linked to our volumetric results, as studies have shown that PFC atrophy and AMYG enlargement are associated with depression [52]. Unexpectedly, we did not find Met treatment to decrease anxiety-like behaviour, which has been seen in previous literature demonstrating that AAS treatment has anxiolytic effects [53]. In line with prior research, we also found that RmTBI decreased total distance travelled in the open field [21]. Additionally, we found no change in general activity due to Met treatment, which agrees with previous studies showing that AAS does not affect general locomotor activity [54]. Of importance, and contrary to previous literature [15,55,56], we failed to identify increased aggression in Met-treated animals in the dominance tube; however, we did see a decreased time in the tube for those rats that received RmTBIs. The literature is lacking on whether this is a dominance or submissive behaviour; however, one study has shown AAS-treated, adolescent rats have heightened sensitivity to provocation [57], which could explain their faster match times.

The PFC plays a vital role in social behaviour, emotional regulation, impulse control, and short-term working memory [42]. We examined mRNA expression changes in the PFC in the context of RmTBI and Met treatment (see Figure 4A–D and Table 3). We did not identify RmTBI-induced changes in the PFC, but there were changes due to Met treatment. *Creb* is a transcription factor that has been heavily implicated in mood disorders such as depression [58]. Contrary to previous studies showing increased *Creb* activity in the PFC of depressed individuals [59], we found a decrease of *Creb* expression in our Met-treated animals, suggesting that Met may affect the neural circuitry of depression in a different manner. We also found increased *Gfap* expression by Met treatment. Increases in *Gfap* have been associated with ageing in both human and rat brains, which is alarming given that the animals in the present study were adolescents [60]. These data imply that Met treatment may be implicated in molecular pathways related to accelerated aging/neurodegeneration, providing a possible mechanism for the PFC atrophy we found in AAS exposed animals.

Given that the AMYG is involved in emotional regulation and the innate stress response [20], altered gene expression may contribute to mood instability and anxiety-related disorders. The only gene found to display RmTBI-induced changes in mRNA expression in the AMYG was *Gfap*, where expression was decreased due to injury. Previous studies have shown decreased *Gfap* expression in models of depressive behaviour [61]; however, this was not reflected in our behavioural assessment. We would have expected an increase in *Gfap* expression due to RmTBI more in-line with previous rodent literature [62]. The interaction between Met treatment and RmTBI was driven in part by sham animals, where those exposed to Met had increased *Gfap* expression. Our results coincide with a study that examined the effect of androgens on the AMYG, which found both an increase in *Gfap* expression and overall AMYG volume due to exogenous androgen administration [63]. Interestingly, Met treatment decreased *Bdnf* expression in the AMYG. A previous study has shown submissive animals to have decreased *Bdnf* expression in the AMYG in a rodent model of social aggression. This possibly indicates that our dominance tube match times were lower in Met-treated animals because they displayed more submissive behaviour or were quicker to enact a flight response from confrontation [64]. Furthermore, Met treatment interacted with RmTBI to influence *Bdnf* expression in the AMYG, whereby injury decreased expression in the placebo groups in addition to AAS exposure decreasing expression in sham animals. Altered levels of *Bdnf* in the AMYG have been associated with depression and impairments in fear responses [65]. *Iba1* is a marker of microglial activation and has been linked to chronic stress and the development of anxiety [31]. Curiously, we did not find any increases in *Iba1* expression due to RmTBI in either brain region, which is a commonly reported measure in rodent mTBI models [66,67]. A reason for this could be that we only examined mRNA expression changes in the PFC and AMYG, when it could have been more pronounced in other brain regions. Finally, we found an interaction between Met treatment and RmTBI on *Creb* expression, where we found the Met-treated animals to have decreased expression in the sham animals. This may reflect the role of *Creb* in social memory formation [68] and may have contributed to dominance tube findings identified in Met-exposed animals. Met treatment had significant effects on mRNA expression in the AMYG, revealing possible molecular pathways for AAS to produce AMYG-dependent depressive and social effects (see Figure 4E–H and Table 3). 

Lastly, our study produced other markers of RmTBI, including changes in telomere length (Figure 4I) and serum GH levels (Figure 4K). As previously shown by our laboratory group, RmTBI decreased telomere length in rats [25], which has been associated with various neurodegenerative diseases [69]. We also saw a large increase in serum GH levels in RmTBI animals compared to shams. This result was unexpected as GH deficiency is one of the most common pituitary defects in clinical mTBI populations [39] and has been demonstrated in rodent models of RmTBI. To our knowledge, this study is the first to demonstrate an increase in serum GH levels in such a model [70]. In a future study, it would be worthwhile to examine levels of IGF-1 as well to determine if the GH–IGF-1 axis was disrupted as elevated GH and lowered IGF-1 levels (indicative of peripheral GH resistance) has been shown in acute stress and critical illness [71]. The elevated GH may also be a result of injury-induced neurogenesis, where increased GH activity might be involved in neuro-restorative processes, which has been shown in a rodent model of ischemic injury [72]. 

## 5. Conclusions

In summary, we failed to support our hypothesis and did not identify cumulative effects of RmTBI and AAS treatment, but did reveal negative manifestations of AAS treatment on the developing brain. Volumetric MRI analysis showed PFC atrophy from both steroids and RmTBI and enlargement of the AMYG due to Met treatment. We identified effects of Met treatment on short-term working memory, dominance behaviour, and depressive-like behaviour in addition to RmTBI effects on general activity. We also found that AAS treatment influenced mRNA expression in PFC and AMYG, possibly altering the neural circuitry involved in depression and cognitive function. While there were many important findings in this study, we must consider the limitations as well. First, our lack of cumulative effects of RmTBI and AAS treatment may have been due to inadequate sensitivity in our behavioural measures, producing a floor effect where it was difficult to observe further performance deficits. Our research group has identified similar outcomes when studying the effects of sleep deprivation and RmTBI [73]. Secondly, although we chose to focus on the PFC, AMYG, and CC for their established roles in mTBI and interactions with AAS exposure, we acknowledge that the rest of the brain is involved in these processes. It would be interesting to examine additional brain regions in the future to fully elucidate the effects of AAS exposure on the developing brain and the different neural circuits and mechanisms involved. Given the alarming results from this study, and that RmTBI and AAS abuse prevalence is on the rise, future experiments regarding AAS use and the developing brain are needed. While our pre-clinical findings provide additional evidence that AAS treatment produces numerous detrimental effects on the developing brain, such as brain atrophy, changes to gene expression, and subsequent behavioural changes; similar MRI analysis on human populations is needed to further investigate this significant public health issue in a clinical setting. 

## Figures and Tables

**Figure 1 brainsci-10-00258-f001:**
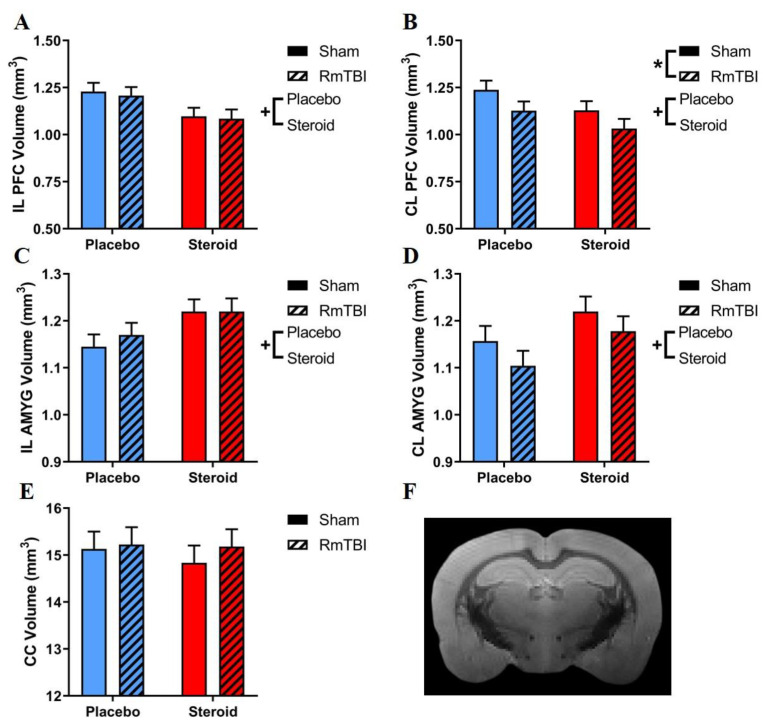
Bar graphs displaying MRI volumetric analysis in regions of interest. (**A**) IL PFC; main effect of steroid treatment whereby steroid groups had lesser volumes than placebo groups. (**B**) CL PFC; main effect of steroid treatment whereby steroid groups had decreased volumes compared to placebo groups; main effect of RmTBI, whereby injured animals had decreased volumes compared to sham animals. (**C**) IL AMYG; main effect of steroid treatment, whereby steroid groups had increased volumes compared to placebo groups. (**D**) CL AMYG; main effect of steroid treatment, whereby steroid groups had increased volumes compared to placebo groups. (**E**) CC; no significant effects from steroid treatment or RmTBI. (**F**) An example of an MRI image. Means ± standard error are displayed where (*) indicates a main effect of RmTBI, and (+) indicates a main effect of steroid treatment; *p* ≤ 0.05. MRI, magnetic resonance imaging; IL, ipsilateral; CL, contralateral; PFC, prefrontal cortex; AMYG; amygdala; CC, corpus callosum; RmTBI, repetitive mild traumatic brain injury.

**Figure 2 brainsci-10-00258-f002:**
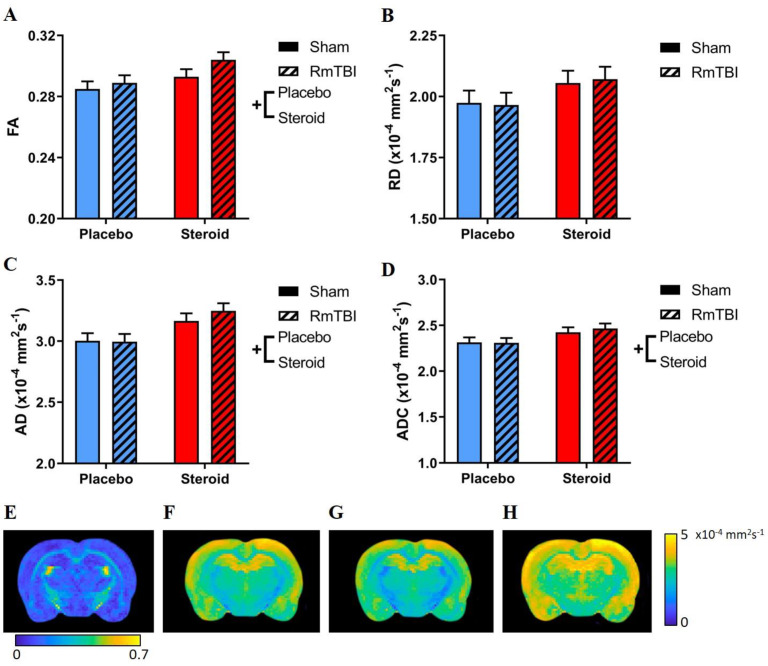
Diffusion-weighted MRI analysis of the CC. Bar graphs displaying DTI metrics. (**A**) FA; main effect of steroid treatment, whereby steroid groups had increased FA compared to placebo groups. (**B**) RD; no effects by steroid treatment or RmTBI. (**C**) AD; main effect by steroid treatment, whereby steroid groups had increased AD compared to placebo groups. (**D**) ADC; main effect of steroid treatment, whereby steroid groups had increased ADC compared to placebo groups. (**E**,**F**,**G**,**H**) displaying example images FA, RD, AD, and ADC, respectively. Means ± standard error are displayed, where (+) indicates a main effect of Met treatment; *p* ≤ 0.05. MRI, magnetic resonance imaging; DTI, diffusion tensor imaging; CC, corpus callosum; FA, fractional anisotropy; RD, radial diffusivity; AD, axial diffusivity; ADC, apparent diffusion coefficient; RmTBI, repetitive mild traumatic brain injury.

**Figure 3 brainsci-10-00258-f003:**
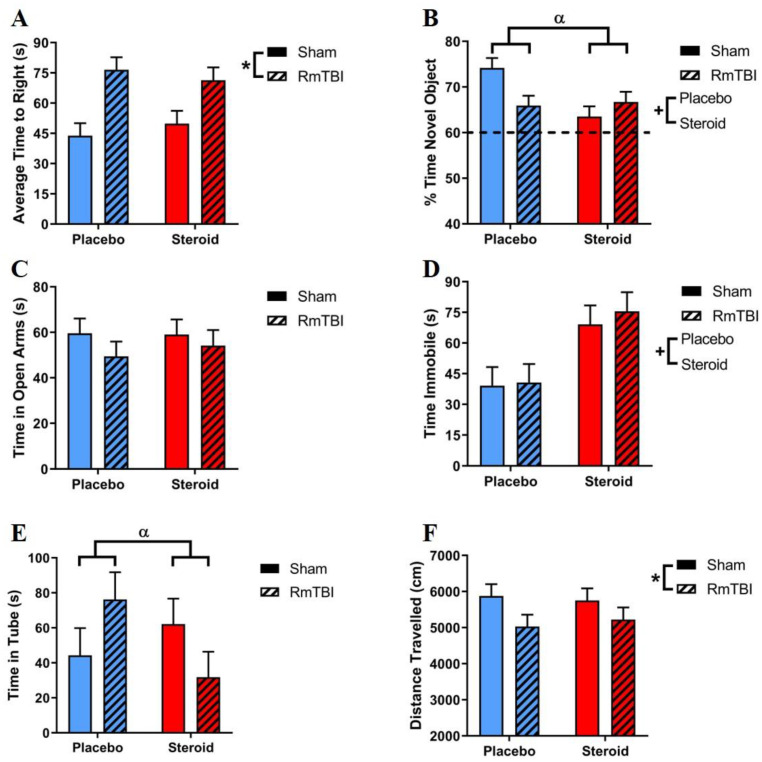
Bar graphs displaying outcomes from a behavioural test battery for all groups. Means ± standard error are displayed, where (*) indicates a main effect of RmTBI, (+) indicates a main effect of Met treatment, and (α) indicates a significant Met treatment by RmTBI interaction; *p* ≤ 0.05. (**A**) displays the average time-to-right after sham injury or RmTBI; main effect of RmTBI, whereby injured animals had increased time-to-right. (**B**) displays the % of time spent with a novel object in the NCM task; main effect of steroid treatment decreasing time spent with the novel object; significant RmTBI by steroid treatment interaction, where Met exposure in sham animals significantly decreased performance. The hatched line indicates the % of expected time the rat will spend investigating the novel object. (**C**) displays the average time spent in the open arms of the elevated plus maze; no main effects of steroid treatment or RmTBI. (**D**) displays the mean time spent immobile in the forced swim test; main effect of steroid treatment, whereby animals in the steroid groups spent significantly more time immobile than placebo groups. (**E**) displays the average time spent in the tube over 3 trials in the dominance tube test; significant RmTBI by steroid treatment interaction driven by injured animals, where the steroid group had lower scores than the placebo group (**F**) displays the mean distance travelled in the open field test; main effect of RmTBI, where injured animals showed decreased activity in the open field. RmTBI, repetitive mild traumatic brain injury; NCM, novel context mismatch.

**Figure 4 brainsci-10-00258-f004:**
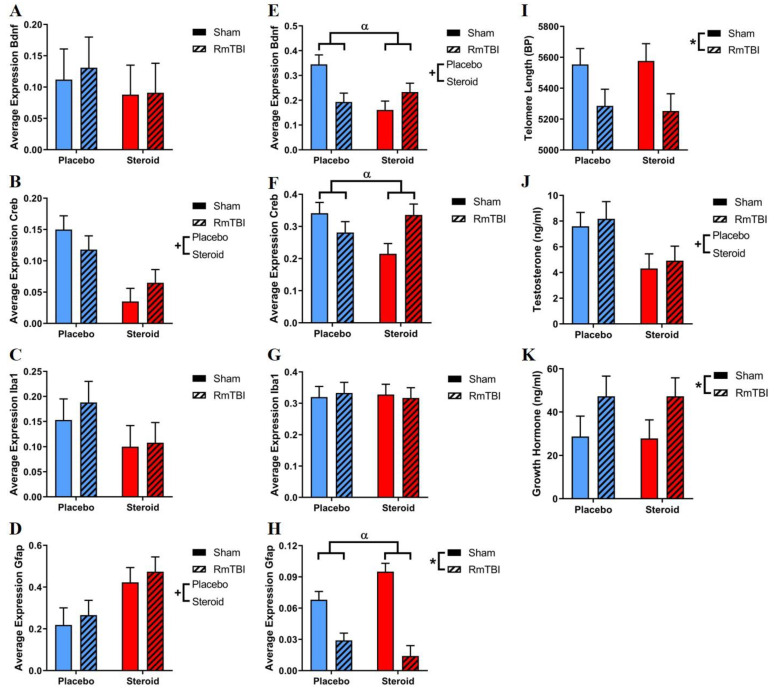
Bar graphs displaying average mRNA expression levels in the PFC and AMYG. (**A**) PFC *Bdnf*; no significant effects. (**B**) PFC *Creb*; main effect of steroid treatment, whereby steroid groups had decreased *Creb* expression (**C**) PFC *Iba1*; no significant effects. (**D**) PFC *Gfap*; main effect of steroid treatment, whereby steroid groups had increased *Gfap* expression. (**E**) AMYG *Bdnf*; main effect of steroid treatment, whereby steroid groups had decreased *Bdnf* expression; significant RmTBI by steroid treatment interaction, whereby steroid exposed sham animals had decreased *Bdnf* expression when compared to sham animals receiving placebo. (**F**) AMYG *Creb*; significant RmTBI by steroid treatment interaction driven by sham animals, whereby steroid treatment decreased *Creb* expression when compared to placebo animals. (**G**) AMYG *Iba1*; no significant effects by steroid treatment or RmTBI. (**H**) AMYG *Gfap*; main effect of RmTBI, where injured animals had decreased expression of *Gfap*; significant RmTBI by steroid treatment interaction driven by the following: (a) the steroid-treated animals, whereby RmTBI decreased *Gfap* expression compared to sham injuries; (b) placebo animals whereby RmTBI decreased *Gfap* expression compared to sham injuries; and (c) by the sham animals, whereby steroid exposure decreased *Gfap* expression compared to placebo-treated animals. (**I**) displays average telomere length at time of euthanasia; main effect of RmTBI, where injured animals had decreased telomere lengths. (**J**) displays average serum testosterone levels at time of euthanasia; main effect of steroid treatment, where steroid-treated animals had lower serum levels of testosterone. (**K**) displays average serum growth hormone levels at time of euthanasia; main effect of RmTBI, where injured animals had increased serum GH levels. Means ± standard error are displayed, where (*) indicates a main effect of RmTBI, (+) indicates a main effect of Met treatment, and (α) indicates a significant steroid treatment by RmTBI interaction; *p* ≤ 0.05. mRNA, messenger RNA; PFC, prefrontal cortex; AMYG, amygdala; RmTBI, repetitive mild traumatic brain injury; BP, base pairs.

**Table 1 brainsci-10-00258-t001:** Statistical results of the two-way ANOVAs for MRI-related outcomes (Volume and DTI) after RmTBI and steroid treatment.

MRI Measure	Brain Region	Effect of RmTBI F (*p*)	Effect of Steroid Treatment F (*p*)	Steroid Treatment x RmTBI F (*p*)
**Volumetric Analysis**	IL PFC	0.12 (0.73)	7.37 (0.01)	0.01 (0.91)
	CL PFC	4.43 (0.05)	4.30 (0.05)	0.02 (0.89)
	IL AMYG	0.22 (0.64)	5.45 (0.03)	0.22 (0.64)
	CL AMYG	2.31 (0.14)	4.62 (0.04)	0.03 (0.86)
	CC	0.35 (0.56)	0.21 (0.65)	0.12 (0.73)
**DTI Measure in the Corpus Callosum**	FA	1.59 (0.22)	4.64 (0.04)	0.38 (0.55)
	RD	0.01 (0.94)	3.43 (0.08)	0.06 (0.81)
	AD	0.36 (0.55)	10.73 (<0.01)	0.49 (0.49)
	ADC	0.10 (0.75)	6.07 (0.02)	0.19 (0.67)

**Table 2 brainsci-10-00258-t002:** Statistical results of the 2-way ANOVAs for the behavioural assessments following RmTBI and steroid treatment.

Behavioural Test	Effect of RmTBI F (*p*)	Effect of Steroid Treatment F (*p*)	Steroid Treatment x RmTBI F (*p*)
Time-to-right	18.74 (<0.01)	0.04 (.95)	0.79 (.38)
Open field: distance	4.23 (0.04)	0.01 (.92)	0.23 (.64)
Open field: time in center	2.20 (0.14)	0.07 (.79)	0.00 (.96)
EPM	1.27 (0.26)	0.10 (.75)	0.16 (.69)
NCM	1.32 (0.25)	5.00 (.03)	6.74 (0.01)
Dominance Tube	<0.01 (0.96)	0.77 (.38)	4.25 (0.04)
Forced Swim	0.19 (0.67)	12.31 (<0.01)	0.07 (0.79)

**Table 3 brainsci-10-00258-t003:** Statistical results of the 2-way ANOVAs for the changes in mRNA expression following RmTBI and steroid treatment.

Brain Region	Gene	Effect of RmTBI F (*p*)	Effect of Steroid Treatment F (*p*)	Steroid Treatment x RmTBI F (*p*)
**PFC**	*Bdnf*	0.06 (0.82)	0.46 (0.50)	0.03 (0.86)
	*Iba1*	0.27 (0.61)	2.51 (0.12)	0.11 (0.74)
	*Creb*	<0.01 (0.99)	15.28 (<0.01)	2.11 (0.16)
	*Gfap*	0.44 (0.52)	7.73 (0.02)	<0.01 (0.98)
**AMYG**	*Bdnf*	1.24 (0.27)	3.94 (0.05)	9.64 (<0.01)
	*Iba1*	<0.01 (0.98)	0.01 (0.91)	0.13 (0.72)
	*Creb*	0.84 (0.37)	1.14 (0.29)	7.44 (0.01)
	*Gfap*	53.11 (<0.01)	0.61 (0.46)	6.45 (0.04)

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
