# Peer review of "Examining the Effects of Anabolic–Androgenic Steroids on Repetitive Mild Traumatic Brain Injury (RmTBI) Outcomes in Adolescent Rats"

_brainsci, 2020, doi:10.3390/brainsci10050258_

Round 1
Reviewer 1 Report
In this manuscript "Examining the Effects of Anabolic Androgenic Steroids and Exercise on Repetitive Mild Traumatic Brain Injury (RmTBI) Outcomes in Adolescent Rats” Jason et.al., reported a valuable insight into the effects of Anabolic-androgenic steroid exposure on adolescent brain development. In this manuscript, the authors used the Rat Repetitive mild traumatic brain injury model to study and analysis of the data. Behavioral testing, MRI, qRT-PCR, and ELISA are the main tools explored for the experimental design.
The comments and suggestions for this manuscript are as follows-
- The authors should provide a more comprehensive introduction and discussion with some more recent published data and appropriate references. Likewise, the conclusion portion for this study needs some improvement.
- Figures 1,2,3 and 4, the image bar graph legends (right corner) are confusing, the author should make it self explanatory including proper statistics. The author must match the bar and respective legend symbols. Likewise, the author can improve the figure legends.
- Page 10, figure3. A large blue rectangle covering the image, the author should take care of this formatting issue.
- The experimental design is appropriate, tables are concise including proper statistical analysis, wherever required.
Author Response
- The authors should provide a more comprehensive introduction and discussion with some more recent published data and appropriate references. Likewise, the conclusion portion for this study needs some improvement.
Response: We have provided a more in-depth coverage of the introduction (see page 1: lines 31-32, 38-42, page 2: 46-47), discussion (see page 12: lines 372-376, 384-387), and to the conclusion (see page 14: lines 550-555)
- Figures 1,2,3 and 4, the image bar graph legends (right corner) are confusing, the author should make it self explanatory including proper statistics. The author must match the bar and respective legend symbols. Likewise, the author can improve the figure legends.
Response: We have updated all figure legends and corresponding references to the bar graphs and believe that they are much more self-explanatory and understandable.
- Page 10, figure3. A large blue rectangle covering the image, the author should take care of this formatting issue.
Response: We have updated the figure files and hope this corrects the issue as there is no blue box on our versions of the documents.
- The experimental design is appropriate, tables are concise including proper statistical analysis, wherever required.
Response: Thank-you.

Reviewer 2 Report
The authors have presented an interesting and well planned evaluation of the effect of steroid consumption in pathophysiology of TBI. With the modifications enclosed below, this work will be an excellent contribution to the foundation of investigating a significant public health interest.
Minor
P 4 Line 127 – Since the authors are evaluating depression as one of the potential sequelae and they are evaluating Bdnf expression levels, they should include a mention of the role of Bdnf in depression with a citation in addition to what they discuss with learning and memory function. It is noted that reference 58 accomplishes this in the discussion section, but this should also be done when the gene is first mentioned.
Figures and tables are referenced in the materials and methods. However, they are not referenced in the discussion section. When the data represented by the figures is discussed in the discussion section, a reference to the figure or table needs to be made.
Line 362 – “….disease or multiple sclerosis [42,43],” The comma at the end of the sentence needs to be a period.
Major
Figure 3 did not render and could not be peer-reviewed. This needs to be fixed for an additional round of peer review to evaluate the claims from these data.
Lines 337-352 An enhanced discussion of the differences in ipsilateral and contralateral volumetric changes in prefrontal cortex is required. In IL PFC, steroid+sham and steroid+TBI are both uniformly lesser in volume than in placebo+sham and placebo+TBI. This is in contrast to CL PFC volumes where we see heterogeneity in the steroid group and the placebo group. In other words, the placebo with sham injury has a greater volume than the placebo plus TBI. And, the steroid plus sham is greater than the steroid plus TBI. Is this a contrecoup effect? The possibility of a contrecoup effect needs to be discussed and further explored beyond “It was unexpected that a lateral TBI produced reductions in the CL PFC and not the IL PFC, contrary to a previous study by our laboratory using the LI technique which found decreases in IL PFC after lateral RmTBIs.”Additionally, both the inter-IL and -CL differences and the intra-IL and -CL differences in each of the four volumes in charts A and B need to be discussed and an attempt to explain why, in IL PFC, the volume of placebo-sham is equal to placebo+TBI and both are greater than sterioid+sham which is equal to steroid+TBI (Table 1 A). And, in contrast CL PFC volume of placebo-sham is greater than placebo+TBI which is equal to sterioid+sham which is greater than steroid+TBI (Table 1 B).
A discussion is also needed of the above general trend in AMYG volumes also have the same volume across the placebo group and the same volume across steroid group in ipsilateral AMYG while the placebo volumes and the steroid volumes are not equivalent across each of the two groups contralaterally (e.g. steroid+sham volume does not equal steroid+TBI volume in contralateral amygdala as compared to ipsilateral amygdala where they are equivalent volumes).
Line 463 – It is not appropriate to generalize this study to adolescent brain development as the duration of this study is orders of magnitude shorter than the duration of brain maturation in the human adolescent, the time course for the initiation of steroid consumption is not age matched for humans (P21 is the usual weaning date for rodents whereas adult males often begin contact sports at 12-16 years; Int. J. Prev. Med. 2013, 4, 624–630) and the study is in a completely different species. This sentence should be changed completely as a PICO analysis would discard this study as having merit for making human clinical decisions or public heath decisions. This study serves as a valuable template for the authors to claim there is evidence warranting performing a similar MRI analysis on human populations, but the claims in this final sentence are not substantiated by the evidence within the study.
Author Response
Minor
- P 4 Line 127 – Since the authors are evaluating depression as one of the potential sequelae and they are evaluatingBdnf expression levels, they should include a mention of the role of Bdnf in depression with a citation in addition to what they discuss with learning and memory function. It is noted that reference 58 accomplishes this in the discussion section, but this should also be done when the gene is first mentioned.
- Response – We have added reference to BDNFs role in depression to the methods section, page 3, line 125.
- Figures and tables are referenced in the materials and methods. However, they are not referenced in the discussion section. When the data represented by the figures is discussed in the discussion section, a reference to the figure or table needs to be made.
- Response – We have added reference to the figures and tables throughout the discussion.
- Line 362 – “….disease or multiple sclerosis [42,43],” The comma at the end of the sentence needs to be a period.
- Response – This has been corrected. Thank-you.
Major
- Figure 3 did not render and could not be peer-reviewed. This needs to be fixed for an additional round of peer review to evaluate the claims from these data.
- Response – We have updated the figures. We are not sure why this was a problem as this was not an issue in our versions of the document. We hope this issue has been resolved.
- Lines 337-352 An enhanced discussion of the differences in ipsilateral and contralateral volumetric changes in prefrontal cortex is required. In IL PFC, steroid+sham and steroid+TBI are both uniformly lesser in volume than in placebo+sham and placebo+TBI. This is in contrast to CL PFC volumes where we see heterogeneity in the steroid group and the placebo group. In other words, the placebo with sham injury has a greater volume than the placebo plus TBI. And, the steroid plus sham is greater than the steroid plus TBI. Is this a contrecoup effect? The possibility of a contrecoup effect needs to be discussed and further explored beyond “It was unexpected that a lateral TBI produced reductions in the CL PFC and not the IL PFC, contrary to a previous study by our laboratory using the LI technique which found decreases in IL PFC after lateral RmTBIs.”Additionally, both the inter-IL and -CL differences and the intra-IL and -CL differences in each of the four volumes in charts A and B need to be discussed and an attempt to explain why, in IL PFC, the volume of placebo-sham is equal to placebo+TBI and both are greater than sterioid+sham which is equal to steroid+TBI (Table 1 A). And, in contrast CL PFC volume of placebo-sham is greater than placebo+TBI which is equal to sterioid+sham which is greater than steroid+TBI (Table 1 B).
- Response – This has been included and updated in the discussion. Please see page 12: lines 365-376.
- A discussion is also needed of the above general trend in AMYG volumes also have the same volume across the placebo group and the same volume across steroid group in ipsilateral AMYG while the placebo volumes and the steroid volumes are not equivalent across each of the two groups contralaterally (e.g. steroid+sham volume does not equal steroid+TBI volume in contralateral amygdala as compared to ipsilateral amygdala where they are equivalent volumes).
- This has also been added; Page 12: line 383-388.
- Line 463 – It is not appropriate to generalize this study to adolescent brain development as the duration of this study is orders of magnitude shorter than the duration of brain maturation in the human adolescent, the time course for the initiation of steroid consumption is not age matched for humans (P21 is the usual weaning date for rodents whereas adult males often begin contact sports at 12-16 years; Int. J. Prev. Med. 2013, 4, 624–630) and the study is in a completely different species. This sentence should be changed completely as a PICO analysis would discard this study as having merit for making human clinical decisions or public heath decisions. This study serves as a valuable template for the authors to claim there is evidence warranting performing a similar MRI analysis on human populations, but the claims in this final sentence are not substantiated by the evidence within the study.
- Response – We have modified the conclusion and feel that we have addressed this comment. See Page 14: line 549-554.

Round 2
Reviewer 1 Report
The author's response is satisfactory.
Reviewer 2 Report
The authors have modified their manuscript in light of the revisions noted. This manuscript is ready to be accepted in its present form.